# Cohort profile: Celiac disease genomic, environmental, microbiome and metabolome study; a prospective longitudinal birth cohort study of children at-risk for celiac disease

Maureen M. Leonard[1,2]*, Victoria Kenyon[2], Francesco Valitutti[3,4], Rita Pennacchio-Harrington[5], Pasqua Piemontese[6], Ruggiero Francavilla[7], Lorenzo Norsa[8], Tiziana Passaro[9], Marco Crocco[10], Mariella Baldassarre[11], Chiara Maria Trovato[12], Alessio Fasano[1,2,3], the CDGEMM working group[¶]

1 Mass General Hospital *for* Children and Division of Pediatric Gastroenterology and Nutrition, Harvard Medical School, Boston, Massachusetts, United States of America, 2 Celiac Research Program, Harvard Medical School, Boston, Massachusetts, United States of America, 3 European Biomedical Research Institute of Salerno (EBRIS), Salerno, Italy, 4 Pediatric Unit, Maternal and Child Health Department, AOU San Giovanni di Dio e Ruggi d'Aragona, Salerno, Italy, 5 Boston College, Boston, Massachusetts, United States of America, 6 NICU, Fondazione IRCCS Ca' Granda Ospedale Maggiore Policlinico, University of Milan, Milan, Italy, 7 Pediatric Unit " Bruno Trambusti", Osp Pediatrico Giovanni XXIII, University of Bari, Bari, Italy, 8 Pediatric Hepatology Gastroenterology and Transplant Unit, Ospedale Papa Giovanni XXIII Bergamo, Bergamo, Italy, 9 Celiac Disease Referral Center, "San Giovanni di Dio e Ruggi d'Aragona" University Hospital, Pole of Cava de' Tirreni, Salerno, Italy, 10 Pediatrics, IRCCS Ospedale Giannina Gaslini, Genova, Italy, 11 Neonatal Intensive Care Unit, University of Bari, Bari, Italy, 12 Celiac Disease Referral Center, Bambino Gesù Hospital, Rome, Italy

☉ These authors contributed equally to this work.
¶ Membership of the CDGEMM working group is provided in the Acknowledgments.
* mleonard7@mgh.harvard.edu

## Abstract

The Celiac Disease Genomic, Environmental, Microbiome and Metabolomic (CDGEMM) study is an international prospective birth cohort in children at-risk of developing celiac disease (CD). The CDGEMM study has been designed to take a multi-omic approach to predicting CD onset in at-risk individuals. Participants are required to have a first-degree family member with biopsy diagnosed CD and must be enrolled prior to the introduction of solid food. Participation involves providing blood and stool samples longitudinally over a period of five years as well as answering questionnaires related to the participant, their family, and environment. Recruitment and data collection have been ongoing since 2014. As of 2022 we have a total of 554 participants and the average age of the cohort is 56.4 months. A total of 54 participants have developed positive antibodies for CD and 31 have confirmed CD. Approximately 80% of the 54 participants with CD have developed it by 3 years of age. To date we have identified several microbial strains, pathways, and metabolites occurring in increased abundance and detected before CD onset, which have previously been linked to autoimmune and inflammatory conditions while others occurred in decreased abundance before CD onset and are known to have anti-inflammatory effects. Our ongoing analysis includes expanding our metagenomic and metabolomic analyses, evaluating environmental

**Data Availability Statement:** All relevant data are within the paper and as a Supporting Information file.

**Funding:** Research reported in this publication was supported by the National Institute of Diabetes and Digestive and Kidney Diseases of the National Institutes of Health under Award Number NIH NIDDK; DK104344 to AF, DK109620 and K23DK122127 to MML. The funders had no role in study design, data collection and analysis, decision to publish, or preparation of the manuscript.

**Competing interests:** The authors have declared that no competing interests exist.

**Abbreviations:** CD, Celiac disease; CDA, Celiac disease autoimmunity; CDGEMM, Celiac Disease Genomic, Environmental, Microbiome and Metabolomic; DGP IgG, Deaminated gliaden protein immunoglobulin G; DOB, Date of birth; EGD, esophagogastroduodenoscopy; EMA, Endomysial antibody; ESPGHAN, European Society for Paediatric Gastroenterology Hepatology and Nutrition; GFD, Gluten free diet; HLA, Human leukocyte antigen; IRB, Institutional Review Board; PBMCs, Peripheral blood mononuclear cells; SNAP, Supplemental nutrition assistance program; tTG IgA, Tissue transglutaminase immunoglobulin A; US, United States; WIC, Women infants and children.

risk factors linked to CD onset, and mechanistic studies investigating how alterations in the microbiome and metabolites may protect against or contribute to CD development.

## Introduction

Celiac disease (CD) is an autoimmune condition characterized by enteropathy and triggered by the ingestion of dietary gluten in genetically predisposed individuals [1]. CD is common, with a global prevalence of 1.4%, in the general population, which varies according to age, sex, and region [2]. Compatible human leukocyte antigen (HLA) genetics are necessary to develop CD but not sufficient as approximately 40% of the population have permissive genetics, but only 3% will develop CD during their lifetime [3]. The incidence of CD is rising in children [4] and adults, and is expected to triple by 2050 [5]. However, neither the genetic predisposition nor the trigger have changed which suggests that other environmental factors are contributing to this rapid rise in incidence. Factors such early life infections [6], infant feeding practices [7], antibiotic exposure [8], and viral pathogens [9] have all been implicated in contributing to CD onset. These factors have also been shown to influence the microbiome [10, 11]. However, to explore these factors, detailed data and sample collection must begin before disease onset [12].

CD is an ideal model to prospectively study autoimmune conditions and the role of environmental factors contributing to CD onset as high-risk children can be identified from birth using HLA genetics [13]. The trigger, gluten, is known and can be accurately traced in terms of timing of introduction and quantity in the diet, and CD onset can be identified by measuring known highly specific autoantibodies in serum [14]. To date three prospective birth cohort studies [13, 15, 16] of children at-risk for CD (due to having genetic compatibility and a family history of CD) have generated information that has improved our understanding of early CD onset in children at increased risk of CD and shaped clinical feeding recommendations [17]. For example, these studies have identified HLA genetics as the strongest risk factor for developing CD [13, 15, 16] and found that neither early gluten introduction [15] at 4 months of age nor late introduction of gluten [13] at 12 months of age affected the risk of CD onset.

The Celiac Disease Genomic, Environmental, Microbiome, and Metabolomic (CDGEMM) study was designed to fill knowledge gaps by focusing on early environmental factors and their interaction with the intestinal microbiome on the earliest steps in CD pathogenesis contributing to the loss of tolerance to gluten and onset of autoimmunity [18]. The specific aims of CDGEMM are (1) To study alterations in the infant's microbiome and metabolome from birth in relation to tolerance *vs*. immune response that leads to the autoimmune insult characteristic of CD; (2) To investigate the impact of specific bacterial strains and bacteria derived metabolites on gut mucosal molecular pathways contributing to CD pathogenesis; and (3) To use multi-omic statistical analysis to identify biomarkers of CD and to predict the chance of CD development. The aim of this cohort profile paper is to provide a comprehensive description of the CDGEMM cohort as a resource for the scientific community to access data and potentially establish collaborative efforts. It includes an overview of the study design, description of the baseline characteristics, and summary of results to date.

## Cohort description

### Study design and study population

The CDGEMM cohort was established to collect prospective, longitudinal, detailed metadata and samples for multi-omic analysis. The CDGEMM cohort recruits pregnant mothers after

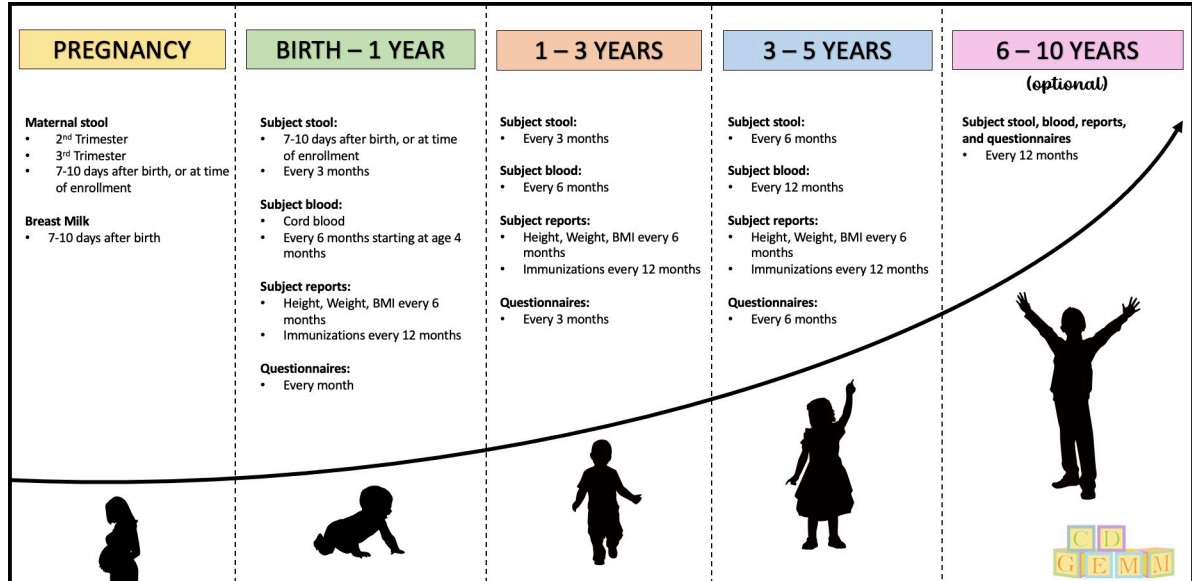

**Fig 1. Study timeline.** Samples and data collected and their frequency are shown.

the first trimester and infants up to 6 months of age from the United States and Italy. Recruited children are characterized as having an increased risk of CD due to having a parent or sibling with CD. Based on previous studies, 75% of these recruited children will have permissive HLA genetics to develop CD and based on their genetics a 10%-26% chance of developing CD by 8 years of age [13, 15].

The study design and sample collection methodologies for the cohort have previously been described [18] and are shown in Fig 1. Briefly, when possible, mothers and infants are enrolled in the study prior to birth to collect samples at delivery and in the first 7–10 days of life. Following this early sample collection, stool is obtained from children every 3 months and blood is collected every 6 months for the first 3 years of life. Stool is collected every 6 months and blood is collected yearly from 3 years of age until completion of the study at either 5 or 10 years of age. Detailed clinical data is obtained regularly as shown in Fig 1 and topics included in the questionnaires are listed in Table 1.

## Ethics

This study was approved by the MassGeneral Brigham Human Research Committee Institutional Review Board (IRB) and the Sapienza University of Rome Comitato Etico (Italian IRB) which serves as the coordinator center in Italy. Parental consent was obtained for each child enrolled.

## Recruitment and retention

### Recruitment

To date, we have recruited 554 participants, 262 (47.5%) in the US and 292 (52.5%) in Italy. Given that we aim to recruit children with a family history of CD prior to birth we required innovative strategies for recruitment of these children. In the US, we employed social media to engage with and identify this unique population. Specifically, we utilized the online community of bloggers and dedicated patient advocacy groups to promote recruitment using Twitter,

**Table 1. Data collection from surveys administered to the parents of children in CDGEMM.**

| Subject and Parental Demographics | Mother's Prenatal Information | Family/Infant Medical History |
|---|---|---|
| Home Address and Email | Pregnancy Weight Gain | First Degree Relative with CD[b] |
| Infants Name, DOB[a], Current Age | Infection Diagnosis | Autoimmune Diseases |
| Parent Name(s) | Antibiotic Use | Mother's Medical Conditions |
| Pediatricians Office | Probiotic Use | Mother's Medications |
| Mother's DOB[a] | Dietary Habits | Child's Medical Conditions |
| Mother's Country of Birth | Assisted Reproductive Services (i.e. IVF) | Child's Symptom Frequency |
| Mother's Educational Level | | Child's Medications |
| Mother and Infant's Race | | Child's Food Allergies |
| Mother and Infant's Ethnicity | | |
| Mother's Height and Weight (before pregnancy) | | |
| **Infant Birth Information** | **Family Structure** | **Family/Infant Dietary Information** |
| Gender | Marital Status | Mother's Dietary Habits |
| Delivery Mode | Infant's Birth Order | Mother's Food Diary and Frequency |
| Weight and Length | Number of Children in the Household | Infant's Food Diary |
| Gestational Age | Number of Adults in the Household | Infant's Feeding Type (Bottle, Breast) |
| Location of Delivery | Pets in the Home | Amount of Milk |
| Hospitalization Post-Delivery | Household Income | Solid Food and Gluten Introduction |
| Neonatal Problems | | Child's Fast Food Consumption |
| Infant Antibiotic Intake | | WIC[c] or SNAP[d] Enrollment |
| Mother's Antiobtioic/Probiotic Intake | | Grocery Store Locations |
| | **Child's Sleep and Activtity** | |
| | Naptime Schedule | |
| | Hours of Sleep | |
| | Hours Watching TV/Video | |
| | Child's Activity Level | |
| | Age Daycare Started | |
| | Daycare Attendance | |

[a] DOB: date of birth.

[b] CD: celiac disease.

[c] WIC: women infants and children.

[d] SNAP: supplemental nutrition assistance program.

Facebook, Instagram, blogs, and podcasts. IRB approved language was provided to all individuals for use on their social media site. A link to our dedicated study website (with both English and Italian translations) and email was provided for interested families to reach out to the study team to either enroll or receive more detailed information about study participation. Social networking both nationally and internationally has been a strong source of recruitment for all sites.

In addition to social media, the study teams in the US and Italy have presented at conferences or expos (related to celiac disease and/or gastroenterology) to reach a local population. Patient referrals from other pediatric gastroenterology providers across the US and Italy have also contributed to successful recruitment. Specifically in Italy, most patients are recruited in clinic or through flyers posted in referring clinic offices. In a short survey asked of all families enrolled reveals social media as being the most successful in the US at 87%, and patient referrals the most prominent for Italy at 80%.

## Retention

Given that this is a longitudinal study which requires serial sample and data collection over the course of 5–10 years, the study team has focused on methods to retain participants and keep both the families and children engaged in the study. Currently, the retention rates are as follows: 82% from the US and 70% from Italy, which can be due to the activities mentioned below.

Upon enrollment into the study, families are provided with a welcome packet that includes:

- Welcome letter from the study team

- Handout with all study team members and contact information

- Handout about the impact the study will have on celiac disease research

- Invitation to join a Facebook group

- Study calendar outlining upcoming visits and what is required (stool, blood, data)

- Progress certificate with spots to add a sticker onto upon completely each year of the study

All sample packages are mailed directly to participants' homes (for blood and stool collection), and the study coordinator schedules bloodwork for the families at a local lab. Additionally, the study team sends out study updates in the form of a newsletter alerting families to new developments or analyses that are ongoing, publications, or sample collection updates. As a follow up to the newsletter, the study team hosts video calls with families, to discuss the study updates and answer any questions they may have.

While strong communication with these participants and their families has been crucial for retention, other creative ways of keeping families involved include:

- A private Facebook group for families to join and get to meet other families, some of whom may be local to them

- A video and book created by the study team designed to outline for the children what the team does with the samples they collect and why they collect them, allowing the children to better understand their participation, especially as they get older

- Each child receives a birthday card with a sticker to place on their progress certificate

- At study completion, each child receives a certificate and personalized lab coat

- For local US families, a party was hosted by the study team in June 2017 and June 2019 at MGH for families to come meet the team and other enrolled participants

The focus of retention techniques has been to ensure each subject feels involved in the study, more than just sending in samples and data, as well as to ensure each child and family understand the impact they are making on CD research.

**Impact of COVID-19.** In March 2020, the CDGEMM study (across all sites) was placed on hold due to the COVID-19 pandemic. All staff members were remote; however, approved study staff were able to package up stool kits to mail to participants to remain on track with specific study time points. Participants were instructed to collect the stool around the appropriate time point and keep in their home freezer until the lab resumed and was back to full operations. In the US, all blood samples were on hold from March to June 2020, and in Italy from March to September 2020; however, upon resuming the blood samples, there were still several time points missed due to participants not able to make it to an outside lab, lab shutdowns, staffing shortages, and several hospitals only allowing urgent cases to be seen, not elective bloodwork. All samples that were missed or delayed were logged appropriately.

The pandemic also led participants to either drop out of the study or become lost to follow up. The most common reason being refusal of study bloodwork. According to records, 9.2% were dropped out of the study from the US and 17.6% from Italy during March 2020 onwards. Additionally, given that participants were not able to obtain bloodwork and upper endoscopy procedures were not available for several months it is possible that there was a delay in CD diagnosis in some participants. In the future we aim to examine whether SARS-CoV-2 has served as a viral trigger of celiac disease and thus increased the incidence of disease in our cohort.

## Findings to date

### Baseline characteristics

The CDGEMM cohort is currently composed of 554 participants enrolled from 42 of the US states, and in 8 different regions in Italy. Baseline demographic data and clinical characteristics are displayed in Table 2. The average age of the overall cohort is 56.7 +/- 23.6 months, with an age range of 0–9 years. Across all sites, we have enrolled 9 sets of twins, and 67 sets of siblings. Additionally, 48% of participants are female and 58% were born via vaginal delivery.

For the purposes of this study, families are allowed to choose whether and/or when they introduce gluten to their child(ren), and thus, the average age at gluten introduction is 8.9 months in the US and 8.1 months in Italy. In the first year of life, parent(s)/guardian(s) report feeding method (breast fed, formula fed, or a combination of both). Of the reported data, in the US, 44.75% of the cohort were primarily breast fed, 15.85% were primarily formula fed, and 10.85% received both breast milk and formula. Moreover, in Italy, 22.83% of the cohort were primarily breast fed, 21.06% were primarily formula fed, and 6.28% received both breast milk and formula. Further data is listed in the S1 Table.

### Samples collected

To date, there have been over 120,000 samples collected and 170,000 data points obtained. There are several different blood sample components that are obtained at each study visit, including: serum, blood clots, whole blood, RNA pax gene, and peripheral blood mononuclear cells (PBMCs). Additionally, at designated sample collection time points stool is collected into both RNA Later solution and without. Data is obtained from parent-reported surveys as well as data obtained from pediatrician offices related to child height, weight, and immunization records. All samples and data listed in Fig 2 are from participants recruited across 42 different states in the US, and 8 regions in Italy.

### Development of CD to date

To date, there are 54 participants (US (n = 17) and Italy (n = 37)) who have had positive tissue transglutaminase IgA (tTG IgA) either through our research lab and/or in a clinical lab. All participants were advised to follow up with local physicians to repeat serology testing and based on those results either confirm the diagnosis with a biopsy or follow ESPGHAN criteria and begin a gluten-free diet without biopsy. Per our study protocol, the following diagnostic criteria is used to categorize all participants who test positive for celiac disease autoantibodies in our research lab:

- *CD by Biopsy*: Upper endoscopy performed revealing Marsh 3 histology.

- *CD by ESPGHAN criteria*: tTG IgA value that is greater than 10 times the upper limit of normal, EMA positive (at least on one occurrence).

**Table 2. CDGEMM subject demographic data.**

|  | USA | ITALY | TOTAL |
|---|---|---|---|
|  | **262** | **292** | **554** |
| **SEX** | | | |
| Female | 128 | 143 | 271 |
| Male | 134 | 140 | 274 |
| **CURRENT AGE** | | | |
| 0-12M | 9 | 12 | 21 |
| 13M-24M | 24 | 25 | 49 |
| 25M-36M | 15 | 27 | 42 |
| 37M-48M | 41 | 59 | 100 |
| 49M-60M | 37 | 36 | 73 |
| >60M | 132 | 130 | 262 |
| **FDR with CD** | | | |
| Mother | 176 | 129 | 305 |
| Father | 35 | 27 | 62 |
| Sibling | 36 | 99 | 135 |
| More than 1 | 12 | 17 | 29 |
| **HLA GENOTYPE** | | | |
| High Risk | 37 | 28 | 65 |
| Standard Risk | 134 | 120 | 254 |
| Low Risk | 37 | 43 | 80 |
| No Risk | 21 | 14 | 35 |
| Unknown or Pending | 33 | 87 | 120 |
| **GESTATIONAL AGE [1]** | | | |
| <37 weeks | 22 | 12 | 34 |
| 37–38 weeks | 48 | 80 | 128 |
| 39–40 weeks | 139 | 119 | 258 |
| >40 weeks | 35 | 36 | 71 |
| **DELIVERY MODE [2]** | | | |
| Vaginal | 182 | 145 | 327 |
| Caesarean section | 78 | 110 | 188 |

[1] Unknown data, n = 63.

[2] Unknown data, n = 39.

- *CD Autoimmunity (CDA)*: tTG IgA value that is positive (or DGP IgG positive if subject is IgA deficient) on at least two occurrences.

- *Potential CD*: tTG IgA value that is > or < 10 times the upper limit of normal, EMA positive, but histology negative.

- *Transient CD*: tTG IgA value that is positive on one occurrence (but not 10 times the upper limit of normal), EMA positive or negative, and then normalization of labs (without adhering to a GFD).

Thus, to date, 16 participants have confirmed CD by biopsy, 15 are considered confirmed CD by ESPGHAN criteria, 9 with CDA, 6 with potential CD, and 8 with transient CD. Of those with transient CD, 4 of them are currently pending clinical workup and the other 4, no symptoms and on a gluten containing diet, are currently being monitored through study lab-work at the set time points. Given that this study is ongoing, participants diagnosed as CDA,

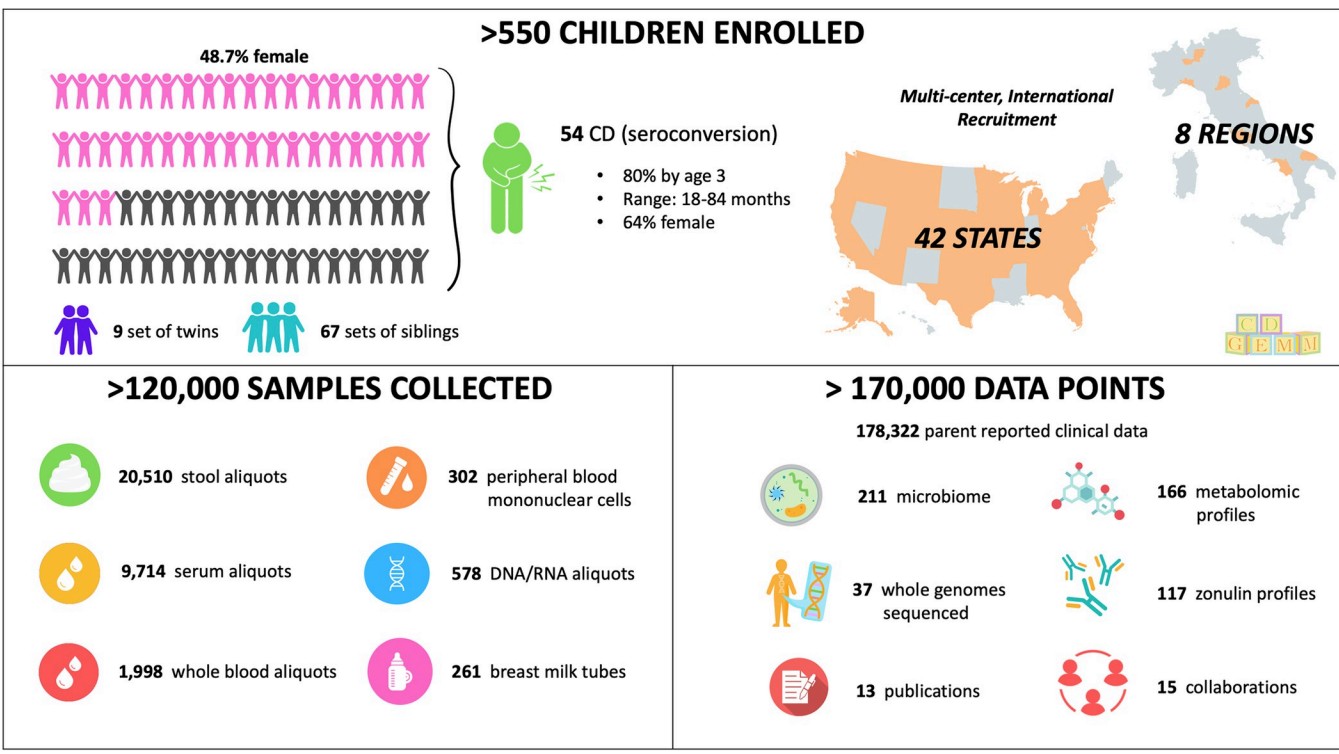

**Fig 2. CDGEMM sample and data overview.** Summary of CDGEMM cohort enrollment. The total number and types of samples collected and data points generated are provided.

Potential CD, and Transient CD are closely followed, unless the participant drops out of the study, with both study and clinical labs.

## Ongoing activities and future plans

We previously published work examining the influence of the HLA genetics and environmental factors including birth delivery mode, infant feeding type, and antibiotic exposure on the gut microbiota composition, function, and metabolome prior to the introduction of gluten [10]. We performed multivariate association, cross-sectional, and longitudinal analyses using metagenomic and metabolomic data collected at three consecutive timepoints (birth, 3 months, and 6 months of age) from 31 children selected from the CDGEMM cohort [10]. Through this work, we ultimately identified taxonomic and functional shifts in the developing gut microbiota of children at risk of CD linking genetic and environmental risk factors to detrimental immunomodulatory and inflammatory effects. After identifying shifts in the microbiota even before the trigger of CD was introduced, we next aimed to identify microbiota signatures predictive of CD onset in children from the CDGEMM cohort who went on to develop CD. Here, we again assessed alterations in the gut microbiota and metabolome before disease onset using shotgun metagenomic sequencing and untargeted metabolomic analysis to perform cross-sectional and longitudinal analyses in the 18 months prior to disease onset in 10 participants that developed CD and the corresponding time point in 10 matched control participants [19]. Our longitudinal analysis identified several microbial species/strains/pathways/metabolites occurring in increased abundance and detected before CD onset which have previously been linked to autoimmune and inflammatory conditions while others occurred in decreased abundance before CD onset and are known to have anti-inflammatory effects. We

also identified previously unreported microbes/pathways/metabolites that could serve as CD-specific biomarkers [19]. We are now in the process of expanding this analysis to include the additional participants that have developed CD.

In other work we utilized a multi-omic approach to examine the microbiota and metabolome of breast milk from mothers enrolled in the CDGEMM study with CD compared to those without CD [20]. We identified differences in bacterial and viral species/strains and in functional pathways between these two groups. We are now expanding these findings in context with the matched infant stool samples from birth to understand how the breast milk microbiota may influence microbiome engraftment in neonates, with potential impact on their future clinical outcomes.

Our next steps include using machine learning to investigate which environmental factors other than gluten may influence the development of CD, taking a mechanistic approach to understanding how microbes identified and isolated from CDGEMM participants may contribute to or protect against the development of CD, and analyzing genomic data, single cell sequencing of isolated immune cells, and epigenetic data. In addition to samples described, we are also generating organoids from duodenal tissue obtained from CDGEMM patients who undergo esophagogastroduodenoscopy (EGD) for suspected CD. The long-term goal of the CDGEMM study is to combine in-depth metadata with multi-omic information inclusive of genomic, microbiome, and metabolomic data and utilize advanced statistical approaches to predict who may develop CD before it occurs to learn how to prevent it.

## Strengths and limitations

The strengths of this cohort include robust metadata and longitudinal sample collection from a well characterized international cohort of children at risk of developing CD. We are predominantly utilizing the fecal microbiome as a surrogate marker of the intestinal microbiome which may be deemed a limitation for the cohort. However, we are obtaining duodenal tissue from participants who undergo EGD which can be used for future analyses including of the intestinal microbiota. In addition, feces must be used as a marker for longitudinal analysis as it is not possible nor ethical to obtain serial endoscopies in otherwise healthy children. Another limitation pertains to the use of self-administered questionnaires which can lead to unreliable and missing data. This limitation is common to longitudinal cohort studies.

In order to understand the earliest steps of CD pathogenesis we must begin detailed data and sample collection and analysis before the onset of disease. The CDGEMM cohort serves as a guide for prospective longitudinal study designs to better understand the role of gut microbiota in disease pathogenesis and to develop therapeutic targets to reestablish tolerance and/or prevent autoimmunity. We welcome collaborations with those interested in efforts to prevent chronic inflammatory disease.

## Supporting information

**S1 Table. Study data.**
(XLSX)

## Acknowledgments

We would like to thank the CD-GEMM working group: Maria Luisa Forchielli MD, Adelaide Serretiello MS, Corrado Vecchi MS, Gemma Castillejo de Villsante MD, Giorgia Venutolo MS, Monica Montuori MD, Basilio Malamisura MD, Angela Calvi MD, Maria Elena Lionetti

MD, Michela Perrone M, Naire Sansotta MD, Annalisa Morelli MD, Lidia Celeste Raguseo MD, Federica Malerba MD, Luca Elli MD, Fernanda Cristofori MD, and Carlo Catassi (MD).

## Author Contributions

**Conceptualization:** Maureen M. Leonard, Francesco Valitutti, Alessio Fasano.

**Data curation:** Rita Pennacchio-Harrington.

**Funding acquisition:** Alessio Fasano.

**Investigation:** Maureen M. Leonard, Francesco Valitutti, Alessio Fasano.

**Methodology:** Maureen M. Leonard, Alessio Fasano.

**Project administration:** Maureen M. Leonard, Victoria Kenyon, Francesco Valitutti, Pasqua Piemontese, Ruggiero Francavilla, Lorenzo Norsa, Tiziana Passaro, Marco Crocco, Mariella Baldassarre, Chiara Maria Trovato, Alessio Fasano.

**Writing – original draft:** Maureen M. Leonard, Victoria Kenyon.

**Writing – review & editing:** Francesco Valitutti, Rita Pennacchio-Harrington, Pasqua Piemontese, Ruggiero Francavilla, Lorenzo Norsa, Tiziana Passaro, Marco Crocco, Mariella Baldassarre, Chiara Maria Trovato, Alessio Fasano.

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
