## [Decision Letter · Decision Letter 0]

8 Feb 2023

PONE-D-23-00282Cohort Profile: Celiac Disease Genomic, Environmental, Microbiome and Metabolome Study; a prospective longitudinal birth cohort study of children at-risk for celiac diseasePLOS ONE

Dear Dr. Leonard,

Thank you for submitting your manuscript to PLOS ONE. After careful consideration, we feel that it has merit but does not fully meet PLOS ONE’s publication criteria as it currently stands. Therefore, we invite you to submit a revised version of the manuscript that addresses the points raised during the review process.

Both reviewers noted a number of minor issues that must be adequately addressed in a revised manuscript.Please submit your revised manuscript by Mar 25 2023 11:59PM. If you will need more time than this to complete your revisions, please reply to this message or contact the journal office at plosone@plos.org. Please include the following items when submitting your revised manuscript:A rebuttal letter that responds to each point raised by the academic editor and reviewer(s). You should upload this letter as a separate file labeled 'Response to Reviewers'.A marked-up copy of your manuscript that highlights changes made to the original version. You should upload this as a separate file labeled 'Revised Manuscript with Track Changes'.An unmarked version of your revised paper without tracked changes. You should upload this as a separate file labeled 'Manuscript'.If applicable, we recommend that you deposit your laboratory protocols in protocols.io to enhance the reproducibility of your results. Protocols.io assigns your protocol its own identifier (DOI) so that it can be cited independently in the future. For instructions see: https://journals.plos.org/plosone/s/submission-guidelines#loc-laboratory-protocols. Additionally, PLOS ONE offers an option for publishing peer-reviewed Lab Protocol articles, which describe protocols hosted on protocols.io. Read more information on sharing protocols at https://plos.org/protocols?utm_medium=editorial-email&utm_source=authorletters&utm_campaign=protocols.

We look forward to receiving your revised manuscript.

Kind regards,

Brenda A Wilson, Ph.D.

Academic Editor

PLOS ONE

Journal Requirements:

4. PLOS requires an ORCID iD for the corresponding author in Editorial Manager on papers submitted after December 6th, 2016. Please ensure that you have an ORCID iD and that it is validated in Editorial Manager. To do this, go to ‘Update my Information’ (in the upper left-hand corner of the main menu), and click on the Fetch/Validate link next to the ORCID field. This will take you to the ORCID site and allow you to create a new iD or authenticate a pre-existing iD in Editorial Manager. Please see the following video for instructions on linking an ORCID iD to your Editorial Manager account: https://www.youtube.com/watch?v=_xcclfuvtxQ.

5. One of the noted authors is a group or consortium CDGEMM working Group. In addition to naming the author group, please list the individual authors and affiliations within this group in the acknowledgments section of your manuscript. Please also indicate clearly a lead author for this group along with a contact email address.

Reviewers' comments:

Reviewer's Responses to Questions

**Comments to the Author**

1. Is the manuscript technically sound, and do the data support the conclusions?

Reviewer #1: Yes

Reviewer #2: Yes

2. Has the statistical analysis been performed appropriately and rigorously? 

Reviewer #1: N/A

Reviewer #2: N/A

3. Have the authors made all data underlying the findings in their manuscript fully available?

Reviewer #1: Yes

Reviewer #2: Yes

4. Is the manuscript presented in an intelligible fashion and written in standard English?

Reviewer #1: Yes

Reviewer #2: Yes

5. Review Comments to the Author

Reviewer #1: An interesting and well written cohort profile manuscript re the development of CD from birth in an at risk population of new births from two geographical areas.

1. Given the high rate of CD development by age 3 years, what proportion of cases correlated with an early or late-on transition from breast milk / formula / or both to solid foods irrespective of gluten introduction? That is was the introduction of solid foods on a similar time-line for all the cases irrespective of geographical location of the participants? Clarify.

2. Also from 1 above and a related query. Are the authors recording the types of overall foods and gluten consumed by the participants? As this reviewer understands it unless gluten content of foods has changed...the expectation is that the majority of wheat grown in the US tends to be high in protein content this being in the form of gluten, whereas in Europe where the majority of wheat grown has lower levels of proteins and as a consequence has a decreased gluten content. Clarify.

3. I agree with the authors that questionnaire data can be difficult to assess in terms of reliable parent reporting and missing data.

4. As a final comment I would like to add that employing the intestinal microbiome as a surrogate marker may have limitations, especially in the absence of samples from healthy children, yet this data could provide information on the progression of intestinal dysbiosis and possible deficits in SCFAs cross feeding that occurs between microbes to trigger pro-inflammatory actions in CD.

Reviewer #2: This manuscript is an update of the status of the CDGEMM cohort study that summarizes the current number of patients and their characteristics, and reviews the scientific papers previously published from the data. The methodology employed to achieve their successful cohort is presented in accessible language and abundant detail, such that others seeking to create patient cohorts could adopt aspects of this approach. The success of the effort to date is clear. Three publications deriving from collected data are described as well as the future investigative agenda. The authors are to be congratulated for achieving longitudinal retention of upwards of 82% of families over 5-10 years. The stated purpose of this descriptive presentation is to invite collaborations. Therefore, the detailed presentation of available samples and data is appropriate.

Minor comments:

Abstract: I think you mean 80% of the 31 who developed CD developed it by age 3. Right now, it reads as if 80% of all participants (554) developed CD.

Development of CD: Is there any speculation as to why more children in the Italian cohort (37 vs 17 US) developed positive TTG? Is that a statistically significant difference? Is it related to specific HLA type?

Can you speculate as to the reason and meaning of transient TTG positivity that then disappears? More details would be interesting (ages, duration of positivity, etc.).

Please provide a reference for the paragraph on page 12 line 290 “In other work . . . .”

Do you want to be more specific in your goal of inviting collaborators into the work? I have not seen this discussed in a scientific paper before, but you may want to end with a sentence on that since it is the stated goal of this descriptive paper.

6. PLOS authors have the option to publish the peer review history of their article (what does this mean?). If published, this will include your full peer review and any attached files.

Reviewer #1: **Yes: **Luis Vitetta

Reviewer #2: No

---

## [Author Response · Author response to Decision Letter 0]

20 Feb 2023

Responses to Reviewer’s Questions

Reviewer #1: An interesting and well written cohort profile manuscript re the development of CD from birth in an at risk population of new births from two geographical areas.

1. Given the high rate of CD development by age 3 years, what proportion of cases correlated with an early or late-on transition from breast milk / formula / or both to solid foods irrespective of gluten introduction? That is was the introduction of solid foods on a similar time-line for all the cases irrespective of geographical location of the participants? Clarify.

We thank the reviewer of this comment. An analysis of dietary factors that may contribute to CD onset was outside the scope of this paper. We are currently analyzing the data you mention as part of a larger study examining dietary and environmental factors in the first 15 months of age that may influence CD onset.

2. Also from 1 above and a related query. Are the authors recording the types of overall foods and gluten consumed by the participants? As this reviewer understands it unless gluten content of foods has changed...the expectation is that the majority of wheat grown in the US tends to be high in protein content this being in the form of gluten, whereas in Europe where the majority of wheat grown has lower levels of proteins and as a consequence has a decreased gluten content. Clarify.

We are recording the types of foods and types of gluten consumed to try to understand how the amount and type may influence CD onset. We agree that there may be differences in wheat protein content between the US and Europe and will explore this in future studies.

3. I agree with the authors that questionnaire data can be difficult to assess in terms of reliable parent reporting and missing data.

Thank you for this comment. 

4. As a final comment I would like to add that employing the intestinal microbiome as a surrogate marker may have limitations, especially in the absence of samples from healthy children, yet this data could provide information on the progression of intestinal dysbiosis and possible deficits in SCFAs cross feeding that occurs between microbes to trigger pro-inflammatory actions in CD.

We agree with your statement. We hope this cohort profile will help us to develop relationships with collaborators who have samples from healthy children which may be used in order to better understand our findings. We have added the following sentence to the end of the paper “We welcome collaborations with those interested in efforts to prevent chronic inflammatory disease.”

Reviewer #2: This manuscript is an update of the status of the CDGEMM cohort study that summarizes the current number of patients and their characteristics, and reviews the scientific papers previously published from the data. The methodology employed to achieve their successful cohort is presented in accessible language and abundant detail, such that others seeking to create patient cohorts could adopt aspects of this approach. The success of the effort to date is clear. Three publications deriving from collected data are described as well as the future investigative agenda. The authors are to be congratulated for achieving longitudinal retention of upwards of 82% of families over 5-10 years. The stated purpose of this descriptive presentation is to invite collaborations. Therefore, the detailed presentation of available samples and data is appropriate.

We thank the reviewer for these comments.

Minor comments:

1. Abstract: I think you mean 80% of the 31 who developed CD developed it by age 3. Right now, it reads as if 80% of all participants (554) developed CD.

We have updated the abstract to read: “Approximately 80% of the 54 participants with CD have developed it by 3 years of age.

2. Development of CD: Is there any speculation as to why more children in the Italian cohort (37 vs 17 US) developed positive TTG? Is that a statistically significant difference? Is it related to specific HLA type?

We are also intrigued by this result as children from the US and Italy all have an increased risk of developing CD due to the family history of CD yet as of now there is a higher incidence in Italy. It is a statistically significant difference. We have not identified why more children in Italy have developed CD. It will be interesting to see once the study is complete if this difference remains. We are currently analyzing environmental and dietary factors as well as looking at serum markers that may contribute to this difference. These studies are ongoing. Our completed studies to date did not identify differences in the microbiome of subjects in the US compared to Italy.

3.Can you speculate as to the reason and meaning of transient TTG positivity that then disappears? More details would be interesting (ages, duration of positivity, etc.).

We have provided data about these subjects with transient CD in the supplement. As stated in the paper 4 of the 8 subjects are undergoing work-up for CD, while the other 4 subjects are still on a gluten-containing diet, no symptoms and tTG IgA has not since been elevated except for that one instance. Since the study is ongoing, we expect some of these subjects will have CD. However, transient tTG IgA elevations have been described in other prospective cohort studies of children at risk of CD. For example, Lionetti et al developed a cohort of 832 children at-risk for CD and described a total of 5 children with low titer tTG IgA from their cohort and 26 children with potential CD that had different outcomes such as development of CD, normalization of antibodies on a gluten containing diet, or fluctuating tTG. (PMID: 25271602). Vriezinga et al followed a cohort of 963 children at risk for celiac disease and describe 8 subjects with transient elevats of celiac-disease associated antibodies for which a diagnosis was never confirmed. (PMID: 25271603). Thus are findings are in line with previous work. 

4. Please provide a reference for the paragraph on page 12 line 290 “In other work . . . .”

Thank you for pointing out this omission. We have provided a reference within the manuscript, which is also below:

Olshan KL, Zomorrodi AR, Pujolassos M, et al. Microbiota and Metabolomic Patterns in the Breast Milk of Subjects with Celiac Disease on a Gluten-Free Diet. Nutrients. 2021;13(7).

5. Do you want to be more specific in your goal of inviting collaborators into the work? I have not seen this discussed in a scientific paper before, but you may want to end with a sentence on that since it is the stated goal of this descriptive paper.

Thank you for this comment. We have added the following to the end of the paper “We welcome collaborations with those interested in efforts to prevent chronic inflammatory disease.”

---

## [Editor Report · Decision Letter 1]

22 Feb 2023

Cohort Profile: Celiac Disease Genomic, Environmental, Microbiome and Metabolome Study; a prospective longitudinal birth cohort study of children at-risk for celiac disease

PONE-D-23-00282R1

Dear Dr. Leonard,

We’re pleased to inform you that your manuscript has been judged scientifically suitable for publication and will be formally accepted for publication once it meets all outstanding technical requirements.

Kind regards,

Brenda A Wilson, Ph.D.

Academic Editor

PLOS ONE
---

## [Editor Report · Acceptance letter]

27 Feb 2023

PONE-D-23-00282R1 

Cohort Profile: Celiac Disease Genomic, Environmental, Microbiome and Metabolome Study; a prospective longitudinal birth cohort study of children at-risk for celiac disease 

Dear Dr. Leonard:

I'm pleased to inform you that your manuscript has been deemed suitable for publication in PLOS ONE. Congratulations! Your manuscript is now with our production department. 

Kind regards, 

on behalf of

Dr. Brenda A Wilson 

Academic Editor

PLOS ONE